# Keypoint Counting Classifiers: Turning Vision Transformers into Self-Explainable Models Without Training

## Abstract

Current approaches for designing self-explainable models (SEMs) require complicated training procedures and specific architectures which makes them impractical. With the advance of general purpose foundation models based on Vision Transformers (ViTs), this impracticability becomes even more problematic. Therefore, new methods are necessary to provide transparency and reliability to ViT-based foundation models. In this work, we present a new method for turning any well-trained ViT-based model into a SEM without retraining, which we call Keypoint Counting Classifiers (KCCs). Recent works have shown that ViTs can automatically identify matching keypoints between images with high precision, and we build on these results to create an easily interpretable decision process that is inherently visualizable in the input. We perform an extensive evaluation which show that KCCs improve the human-machine communication compared to recent baselines. We believe that KCCs constitute an important step towards making ViT-based foundation models more transparent and reliable. Code is available at *anonymised* and in the supplementary.

## 1 Introduction

Vision Transformer (ViT)-based (Dosovitskiy et al., 2021) foundation models are becoming increasingly important in computer vision (Han et al., 2023), but are still limited in safety critical domains due to their lack of explainability (Longo et al., 2024). Self-explainable models (SEMs) (Chen et al., 2019; Gautam et al., 2022; Bassan et al., 2025) provide a promising direction within explainable artificial intelligence (XAI) that can address this lack of explainability. SEMs are a type of algorithm where the decision process is inherently explainable, which is highly important to address the disagreement problem (Krishna et al., 2024) that standard post-hoc explainability methods suffer from. However, SEMs have several limitations that limit their usability (Hoffmann et al., 2021), and here we highlight two key limitations:

**(1) SEMs lack flexibility**   A highly contributing factor to the usefulness of ViT-based foundation models is their high flexibility and generalization capabilities. Often, they can be applied directly to a new tasks without the need for finetuning or adjustments. However, SEMs regularly assume a particular architecture, for example a convolutional neural network (CNN)-based feature extractor (Chen et al., 2019; Nauta et al., 2023), which makes them incompatible with ViT-based foundation models. Moreover, SEMs that do not have these constraints, often require an additional classification head to be trained on top of the ViT (Turbé et al., 2024). This requires additional training, which reduces the flexibility of the entire systems. Therefore, there is a need for new methodology that can turn ViT-based foundation models into SEMs while keeping their flexibility.

**(2) Current SEMs visualize explanations poorly**   In computer vision, the explanations of SEMs are either presented as bounding boxes (Nauta et al., 2023) or heatmaps (Gautam et al., 2024). These two approaches are both standard methods in SEMs, despite the fact that many studies have pointed out limitations associated with both bounding boxes and heatmaps. For bounding boxes, the construction of the bounding boxes entails finding a rectangle that contains a certain amount of relevance. Several works have pointed out that these bounding boxes can be misleading and cover

more area than the actual activation site (Hoffmann et al., 2021; Davoodi et al., 2023). Heatmaps are usually overlaid on an image to highlight what region are important for a decision, and have been criticized for lacking precision (Carmichael et al., 2024), being uninformative (Xu-Darme et al., 2023), and having low performance in user studies (Kim et al., 2023). To improve the usability of SEMs, new methods for visualizing explanations must be developed.

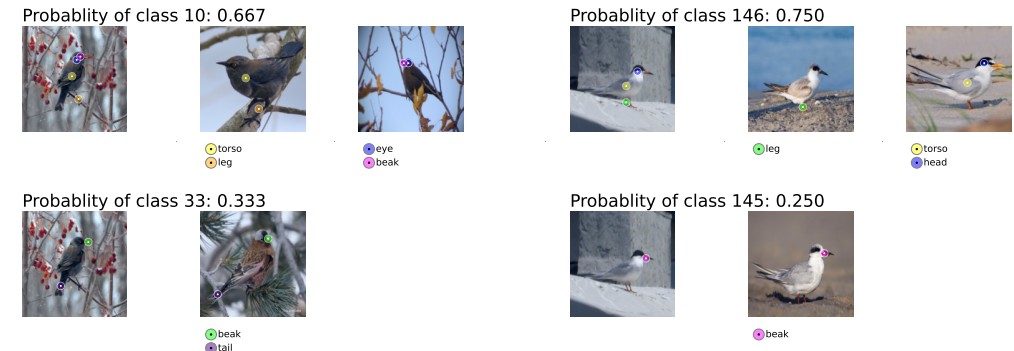

Figure 1: A demonstration of KCCs in the context of bird classification. KCCs identify matching keypoints between a query (leftmost image) and a set of prototypes (rightmost images). Only prototypes with matches are shown to avoid overloading the reader. Predictions are made by counting the number of matches. Here, we leverage ViTs with vision-language capabilities to automatically describe the keypoints (Sun et al., 2023). Note that the class names in the predictions are deliberately omitted to avoid readers using the class names instead of the explanation Kim et al. (2022).

In this work, we address these limitation through a novel framework called Keypoint Counting Classifiers (KCCs), which can turn any ViT-based foundation model into a SEM without the need to train the feature extractor or an additional classification head. KCCs compare *parts* of the query image with *parts* from the prototypes, which is accomplished by exploiting the part-correspondence in the tokens of ViTs (Amir et al., 2022). A set of matching regions between the query and the prototypes is identified through the use of mutual NNs (Oron et al., 2018), and the final classification decision is reached by counting matches between the query and the prototypes. This procedure is performed without any retraining, thus addressing the lack of flexibility. Then, the explanations are visualized as matching keypoints, which is a completely new approach for visualizing explanation that aims to improve the communication of the explanations.

The use of keypoints to visualize explanations is motivated by its frequent use in teaching material and visual learning. One example is in the study of birds, where keypoints are commonly used to visualize critical parts of the animal, which even has its own name (bird topography (Pettingill, 1984)). Another example is in the study of human anatomy, where keypoints are the standard way to highlight important anatomical features in teaching material (see for example (Netter, 2017)). Studies on how people learn most effectively have highlighted that people can only process a few pieces of information at the time to reduce cognitive overload (Mayer, 2018). Our motivation for introducing keypoints is to avoid this overload that uninformative and imprecise heatmaps and bounding boxes could introduce. Additionally, we show how in the special case of ViTs with vision-language capabilities, the keypoints can be automatically labeled, which is motivated from the perspective of reducing reader bias Bove et al. (2024). Figure 1 show two examples if the explanation provided by a KCC in the conext of bird classification. In summary, KCCs constitute a new paradigm among SEMs with increased flexibility and improved visualization of explanations. Our contributions are:

1. We introduce Keypoint Counting Classifiers; a general purpose method for turning ViT-based foundation models into SEMs without any retraining.

2. We conduct an extensive quantitative evaluation with comparison to recent and relevant baselines that demonstrate the benefits of KCCs.

3. We conduct a human user study which shows that KCCs improve human-machine communication compared to existing alternatives.

4. We show how in the particular case of ViTs with vision-language capabiliies, keypoints can be automatically labeled which makes initial progress towards reudcing reader bias.

## 2 RELATED WORK

Creating self-explainable models through prototypical parts is one of the main directions within self-explainability, and is often motivated from the perspective of recognition-of-component theory (Biederman, 1987). The pioneering work of ProtoPNet (Chen et al., 2019) and its numerous derivatives (Donnelly et al., 2022; Rymarczyk et al., 2022; Nauta et al., 2023) paved the way for self-explainable deep learning-based approaches for image classification. The core idea of these approaches is to use a CNN-based encoder that retains some spatial resolution and learn interpretable part detectors that are linearly combined to perform classification. A fundamental limitation of these approaches is that they are constrained to particular CNN-based architectures and can therefore not take advantage of advances in ViT-based foundation models. Recent works have proposed self-explainable models built specifically on ViTs. The ViT-NeT combines a ViT backbone with a tree-based classifier (Kim et al.), and the ProtoS-ViT combined a ViT backbone with additional trainable layers and a compactness regularization (Turbé et al., 2024) . However, both of these approaches require an additional training step that limits their flexibility. Some notable works can turn pre-trained models into self-explainable models. The TesNet provides a plug-in transparent embedding space (Wang et al., 2021), but is restricted to work on CNNs. More recently, the KMEx framework can turn any pretrained model into a SEM (Gautam et al., 2024). This is accomplished by creating a 1-nearest-neighbor (NN) classifier in combination with feature visualization techniques. However, the 1-NN setup restricts the explanation, as only a single image can be visualized as the basis of the explanation. Thus, there is a need for new training-free methodology with greater flexibility.

## 3 KEYPOINT COUNTING CLASSIFIER - TURNING VIT-BASED FOUNDATION MODELS INTO SEMS

We present a new paradigm for SEMs, a keypoint-based approach that requires no additional training and a new way of visualizing explanations. Our motivation for introducing KCCs is the high performance and flexibility of ViT-based foundation models in classification and keypoint matching without additional training (Oquab et al., 2024; Amir et al., 2022) and the frequent use of keypoints in teaching and visual learning (Pettingill, 1984; Netter, 2017). KKCs are constructed through three parts: (1) an image-wise identification of keypoints, (2) keypoint matching through mutual NNs, and (3) classification through counting matching keypoints. These steps are described below.

**Preliminaries on ViTs** Let $\mathbf{X} \in \mathbb{R}^{C \times H \times W}$ denote an image with $C$ channels, a width of $W$ pixels and a height of $H$ pixels that is reshaped into a sequence of flattened 2D patches $\mathbf{X}^{(p)} \in \mathbb{R}^{N_T \times P^2}$, where $N_T = HW/P$ and $P$ is the height and width of each image patch. A ViT transforms $\mathbf{X}^{(p)}$ into a set of latent representations, referred to as tokens, $Z = \{\mathbf{z}_1, \cdots, \mathbf{z}_{N_T}\}$ and a classification token $\mathbf{z}_{cls}$ that summarizes the information across all tokens into a single representation. All tokens are of the same dimensionality $D$. A positional embedding is added to each patch embedding to retain positional information from the input space. Due to the positional embeddings, the set of tokens can also be rearranged into a low resolution encoded representation of the input $\mathbf{Z} \in \mathbb{R}^{\sqrt{N_T} \times \sqrt{N_T} \times D}$.

### 3.1 PART 1 - IMAGE-WISE IDENTIFICATION OF KEYPOINTS

The fundamental idea of KCCs is to count matching keypoints between a query and a set of prototypes. The first part of KCCs is therefore to identify keypoints in the query and all prototypes. A recent work by Amir et al. (2022) showed that the tokens of ViT capture information about semantic parts, and we will build on these findings to identify keypoints.

First, we identify foreground pixels in $\mathbf{X}$. This is necessary since tokens also can capture information about the background, which should not be included when matching keypoints between objects. Formally, let $f$ be a function that takes in an image $\mathbf{X}$ and outputs a binary foreground mask $\mathbf{F} \in \{0, 1\}^{H \times W}$, where 1 indicates foreground pixels and 0 indicates background pixels. Foreground identification is a fundamental problem in computer vision that can be handled in many ways, and here we treat the selection of $f$ as a task-specific hyperparameter. In some cases, privileged information in the form of segmentation masks could be available. When privileged information is not available, general purpose foundation models have been shown to provide excellent

foreground segmentation across a diverse set of visual tasks (Ren et al., 2024). If external models are not accessible, the ViT itself can be used to identify foreground pixels through the attention mechanism (Amir et al., 2022) or through subspace analysis of the tokens (Oquab et al., 2024). In this work, we rely on the combination of the Segment Anything model (Kirillov et al., 2023) and Grounding DINO to perform the foreground segmentation, due to its impressive performance across numerous diverse and challenging tasks (Ren et al., 2024).

Second, the foreground is split into semantically coherent segments. These segments will form the basis for the keypoints and is computed based on the tokens due to their strong alignment with object parts (Amir et al., 2022) through the $\mathbf{Z}$ representation of the image $\mathbf{X}$. A challenge when operating on $\mathbf{Z}$ is that the resolution is much lower than the input and not compatible with the foreground mask $\mathbf{F}$. We found that a simple solution to this challenge was to downsample $\mathbf{F}$ using NN interpolation and upsample $Z$ using bilinear interpolation to a new resolution $H' \times W'$ between $H \times W$ and $\sqrt{N_T} \times \sqrt{N_T}$, was a good compromise between reducing computational demand and avoiding overly coarse part-segments.

Mathematically, let $s$ be a function that takes in the upsampled token representations $\mathbf{Z}$ and outputs a discrete mask $\mathbf{S} \in \{0, \cdots, N_s\}^{H' \times W'}$, where each integer indicates membership to a semantically coherent segment and $N_s$ is the number of segments for that particular image. Again, we treat this process as a task-specific hyperparameter. If available, privileged information about part locations can be used, and if not, general purpose methods based on traditional image processing techniques or more recent deep learning-based approaches are available (Achanta et al., 2012; Aniraj et al., 2024). Compared to the foreground identification task, part-segmentation is more challenging and current deep learning approaches often must be trained for a particular dataset (Aniraj et al., 2024). Therefore, to retain flexibility and avoid additional training we leverage the widely used SLIC method (Achanta et al., 2012) to separate the foreground into semantically coherent segments due to its reliability and flexibility.

Finally, we define a keypoint as the center of a segment. The entire first part of KCCs is illustrated in columns 1-4 of Figure 2. Since this part of the method is performed image-wise, the entire procedure can be precomputed for the prototype keypoints, which can greatly speed up inference in cases where many prototypes are necessary.

## 3.2 PART 2 - IDENTIFY MATCHING KEYPOINTS

Once the keypoints have been located, the second part of KCCs is to identify matching keypoints. To perform the matching, we take inspiration from template matching using mutual NNs (Oron et al., 2018), which has been demonstrated to effectively locate matching keypoints between two images (Amir et al., 2022).

The mutual NN procedure requires computing similarities between the keypoints. Here, we propose to represent each keypoint as the mean representation of all tokens within a segment of $\mathbf{S}$, and compute similarities between these representations. Formally, we calculate the mean representation of segment $i$ for a given image $\mathbf{X}$ as:

$$\boldsymbol{\mu}_i = \frac{1}{K_i} \sum Z(j), \text{ for all } j \text{ where } S(j) = i, \tag{1}$$

where $K_i$ is the number of tokens belonging to segment $i$. Next, we can determine if two segments are mutual NNs as follows:

$$MNN(\boldsymbol{\mu}_i^q, \boldsymbol{\mu}_j^{p_k}, U^q, U^P) = \begin{cases} 1, & \text{if } NN(\boldsymbol{\mu}_i^q, U^P) = \boldsymbol{\mu}_j^{p_k} \text{ and } NN(\boldsymbol{\mu}_j^{p_k}, U^q) = \boldsymbol{\mu}_i^q \\ 0, & \text{else} \end{cases} . \tag{2}$$

where

$$NN\left(\boldsymbol{\mu}_i^q, U^P\right) = \underset{\boldsymbol{\mu}^P \in U^P}{\arg\min} d\left(\boldsymbol{\mu}_i^q, \boldsymbol{\mu}^P\right) \tag{3}$$

and $d(\cdot, \cdot)$ is some distance measure. In Equation 2, $\boldsymbol{\mu}_i^q$ is the representation of segment $i$ in the query, $\boldsymbol{\mu}_i^{p_k}$ is the representation of segment $j$ in prototype $k$, $U^q$ is the set of all query segment representations, and $U^P$ is the set of all $k$ prototype segment representations. Equation 2 is repeated

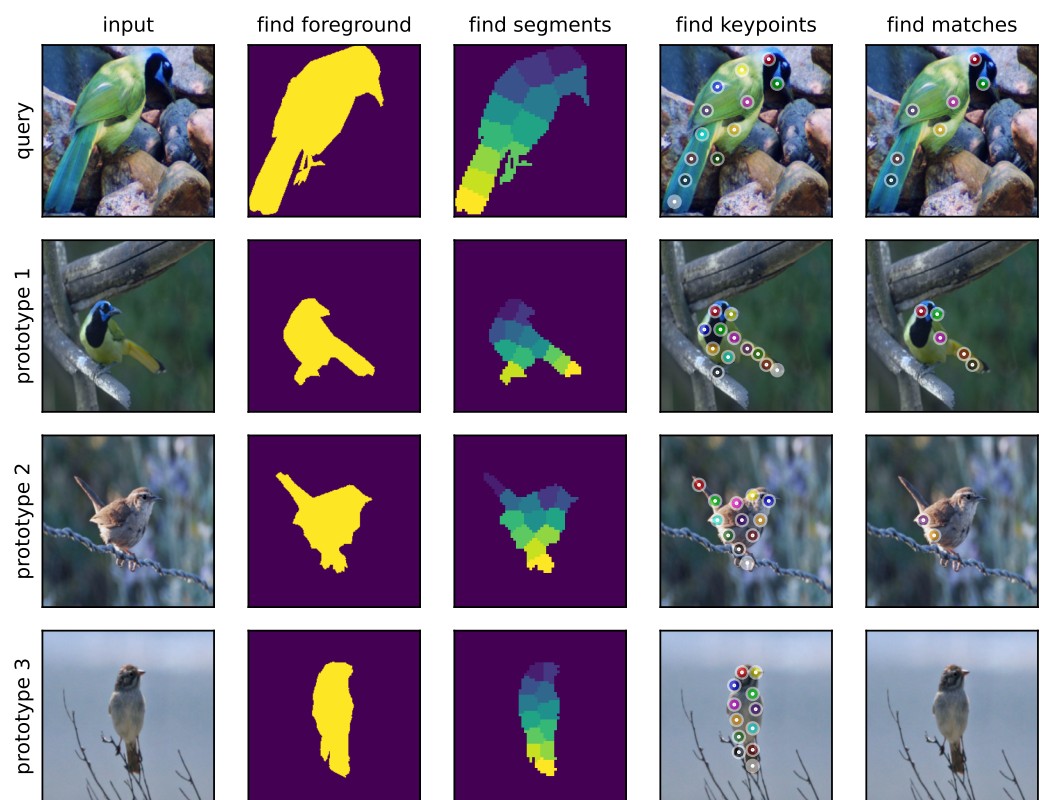

Figure 2: Illustration of each step of KCCs. Going from column 4 to column 5, mutual NNs are computed between the keypoints in the query and all prototype keypoints. Only keypoints that are mutual NNs kept. In this case, prototype 3 has no mutual NNs with the query, and is therefore without keypoints.

across all query and prototype tokens, and tokens that are identified as mutual NNs are collected in the set $M$. Additionally, we create a corresponding set $Y^M$ that contains the class label of the matched prototype. The last column Figure 2 shows the remaining keypoints after the matching.

### 3.3 PART 3 - CLASSIFY BY COUNTING MATCHING KEYPOINTS

After identifying matching keypoints, the final part of KCCs is to classify the query based on the number of matches. For a given class $c$, we count the number of matches between the query and prototypes belonging to the image-wise class $c$ as:

$$\hat{y}_c = \frac{1}{|M|} \sum Y^M(i), \text{ for all } i \text{ where } Y^M = c. \tag{4}$$

## 4 EVALUATION AND EXPERIMENTAL SETUP

We describe how KCCs are evaluated, how we identify prototypes, and how we select and evaluate important hyperparameters in KCCs.

**Evaluation** We follow the established evaluation protocol for SEMs (Chen et al., 2019; Nauta et al., 2023; Rymarczyk et al., 2022; Turbé et al., 2024) and measure accuracy and complexity. Complexity is measured by calculating the average number of images a user must inspect as part of the explanation. We evaluate on widely used fine-grained classification datasets; CUB200 (Wah et al., 2011), Stanford Cars (Krause et al., 2013), and Oxford Pets (Parkhi et al., 2012), and com-

pare with numerous strong baselines; ProtoPNet(Chen et al., 2019), ProtoTree (Nauta et al., 2021), ProtoPool (Rymarczyk et al., 2022), PIP-Net (Nauta et al., 2023), ProtoS-ViT (Turbé et al., 2024), ViT-NeT (Kim et al.), ST-ProtoPNet (Wang et al., 2023), and KMEx (Gautam et al., 2024).

**User study**   We conducted a comprehensive user study to evaluate how users perceive different explanation types generated by SEMs. The study design was based on the HIVE framework, which allows for falsifiable hypothesis testing, cross-method comparison, and human-centered evaluation (Kim et al., 2022). In particular, we included the agreement task and subjective evaluation questions, and excluded any biasing factors like class labels. The study compared three visualization methods: bounding boxes, heatmaps, and our proposed keypoints. Drawing on prior research indicating user preference for example-based and part-based explanations (Kim et al., 2023), we included two baseline methods: PiP-Net (Nauta et al., 2023), which offers prototype-based explanations, and KMEx (Gautam et al., 2024), which uses NN examples (i.e. example-based). PiP-Net explanations were visualized using bounding boxes, whereas KMEX employed heatmaps. Each method was represented by six correct and six incorrect classifications, with participants viewing three randomly selected examples per method. All examples were drawn from the same set of classes across methods to control for class difficulty. The order of methods was randomized between participants. Before the study, all participants received a short lecture explaining all methods to ensure a common understanding of the explanations. We evaluate the *agreement* with the model prediction, the *quality* of the shown explanation and the *understanding* of the explanation as follows:

- *agreement*: How confident are you that the model prediction is correct based on the explanation? (scale 1-4, 1/2: prediction is incorrect, 3/4: prediction is correct)
- *quality*: Do you agree with the key point matches/heatmap/matched patches? (scale 1-4, 1/2: keypoints/heatmap/patches are incorrect, 3/4: keypoints/heatmap/patches are correct)
- *understanding*: How easy is it to understand the explanation? (Scale 1-4, 1: Very difficult, 4: Very easy)

For *agreement* we report how many identified the correct and incorrect predictions correctly. For *quality* and *understanding*, we report the mean answer and standard deviation. Differences in methods are compared with a 2-sample t-test (p-value $< 0.05$ is considered significant).

Overall, 51 participants with medium to high expertise in AI ($3.66\pm0.66$, scale 1-5 (1:no experience, 5:expert)) and low expertise in birds ($1.51\pm0.75$, scale 1-5 (1:no experience, 5:expert)) participated in the study.

**Finding prototypes and hyperparameters**   Following prior works Chen et al. (2019); Gautam et al. (2022), we calculate 10 prototypes per class, and we use the cosine similarity in Equation 3. Note, since similarity is computed between all tokens in the query and all tokens in all prototypes, this can quickly become memory intensive if there are many classes. Therefore, we only consider the distance to tokens in the query to tokens of the $J$ closest prototypes (in terms of cosine similarity). We evaluate the performance of KCCs for several choices of $J$ in the experiments. For the number of segments ($N_s$), we report results for 8 and 12 segments per image.

## 5   RESULTS

We present the main results of our evaluation. First, we show the results of the user study, followed by the performance evaluation in terms of accuracy and complexity. Then, we show how in the particular case of ViT-based foundation models with vision-language capabilities, KCCs can be imbued with automatically generated keypoint descriptors in the form of text to reduce reader bias and enhance user friendliness. In Appendix B, we perform a study of hyperparameters like the number of keypoints and prototypes and evaluate the performance across different ViTs.

**User study**   We conducted a user study to evaluate the three explanation methods with different visualizations: PIP-Net, KMEx, and our proposed method, KCC. The study assessed participants' ability to identify correct and incorrect predictions (*agreement*), perceived explanation *quality*, ease of *understanding*, and overall *user preference*, the main results are shown in Table 1.

Table 1: Results of user study. The table reports user agreement with model predictions (correct and incorrect), perceived explanation quality (mean ± std), ease of understanding, and overall user preference. Bold numbers indicate statistical significance (2-sample t-test, p-value < 0.05). KCC achieved the highest scores in explanation quality and understanding, and was tied with KMEx as the most preferred method. The agreement task highlights the confirmation bias (Skitka et al., 1999); users trust the model's prediction even if it is wrong.

| method | type of visualization | prediction | agreement | quality | understanding | user preference |
|---|---|---|---|---|---|---|
| PIP-Net | bounding box | correct | 77.92% | $3.06 \pm 0.68$ | $2.53 \pm 0.77$ | 17.6% |
| | | incorrect | **46.05%** | $2.70 \pm 0.67$ | $2.51 \pm 0.79$ | |
| KMEx | heatmap | correct | 85.71% | $2.84 \pm 1.00$ | $2.99 \pm 0.90$ | **41.2%** |
| | | incorrect | 35.53% | $2.50 \pm 1.15$ | $2.78 \pm 0.92$ | |
| KCC (ours) | keypoint | correct | **88.16%** | $\mathbf{3.34 \pm 0.62}$ | $\mathbf{3.05 \pm 0.71}$ | **41.2%** |
| | | incorrect | 35.06% | $\mathbf{3.10 \pm 0.82}$ | $\mathbf{2.81 \pm 0.80}$ | |

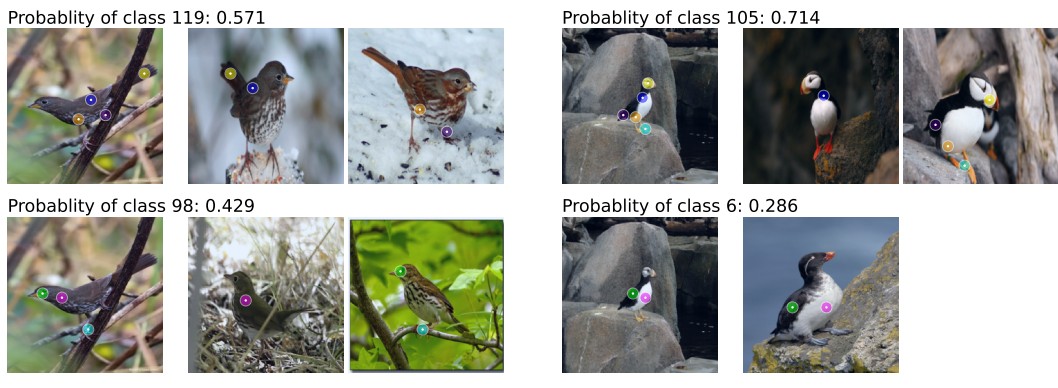

(a) Query (left) correctly classified as Fox Sparrow. Second most probable class is Ovenbird.

(b) Query (left) correctly classified as Horned Puffin. Second most probable class is Parakeet Auklet.

Figure 3: Qualitative examples of KCC explanations.

KCC achieved the highest rate of correct prediction identification (88.16%), significantly outperforming PIP-Net (77.92%, p ¡ 0.05). KMEx also performed well (85.71%), the difference compared to KCC was not statistically significant. For incorrect predictions, PIP-Net had the highest agreement rate (46.05%), but this difference was not statistically significant compared to the other methods. Across all methods, users correctly identified only 35-46% of incorrect predictions, indicating a high degree of confirmation bias. This aligns with previous observations in the HIVE study (Kim et al., 2022), suggesting that users tend to trust model outputs even when they are incorrect when they are shown explanations.

Participants rated KCCs highest in explanation quality ($3.34 \pm 0.62$ for correct predictions), significantly outperforming both KMEx ($2.84 \pm 1.00$) and PIP-Net ($3.06 \pm 0.68$). For incorrect predictions, KCCs again received the highest quality ratings ($3.10 \pm 0.82$), which means that the keypoint matches are mostly perceived as correct, explaining the high trust in the models prediction even if it is wrong. The quality of the heatmaps for KMEx was generally perceived as moderate ($2.50 \pm 1.15$), indicating some ambiguity in interpretation. However, this did not appear to reduce user confidence in the models predictions, potentially due to the intuitive nature of NN examples. In terms of ease of understanding, KCC was rated the most comprehensible ($3.05 \pm 0.71$), significantly better than PIP-Net ($2.53 \pm 0.77$), but not significantly different from KMEx ($2.99 \pm 0.90$).

When asked to indicate their preferred method, users favored KMEx and KCC equally (41.2% each), with PIP-Net receiving the fewest votes (17.6%). This suggests that while KCC offers the best overall interpretability, KMEx remains a strong alternative.

| method | ViT? | no retraining of encoder | no training of clf head | CUB200 | | CARS | | PETS | |
|--------|------|---------------------------|--------------------------|--------|------|------|------|------|------|
| | | | | A | C | A | C | A | C |
| ProtoPNet | ✗ | ✗ | ✗ | 79.2 | 2000 | 86.1 | 1960 | - | - |
| ProtoTree | ✗ | ✗ | ✗ | 82.2 | 8.3 | 86.6 | 8.5 | - | - |
| ProtoPool | ✗ | ✗ | ✗ | 85.5 | 202 | 88.9 | 195 | - | - |
| PIP-Net | ✗ | ✗ | ✗ | 84.3 | 4.0 | 88.2 | 4.0 | 92.0 | 2.0 |
| ProtoS-ViT | ✓ | ✓ | ✗ | 85.2 | 6.0 | 93.5 | 7.0 | 95.2 | 4.0 |
| ViT-NeT | ✓ | ✓ | ✗ | 91.6 | - | 93.6 | - | - | - |
| ST-ProtoPNet | ✓ | ✓ | ✗ | 86.1 | - | 92.7 | - | - | - |
| KMEx | ✓ | ✓ | ✓ | 85.0 | 1.0 | 59.4 | 1.0 | 94.0 | 1.0 |
| KCCs (ours) | ✓ | ✓ | ✓ | 82.2 | 2.2 | 44.8 | 2.3 | 90.0 | 2.3 |

Table 2: Evaluation of accuracy and complexity of numerous SEMs across three widely used benchmark datasets. Due to architectural constraints and training vs. no-training, a direct comparison is challenging, but all ViT-based model in this table are based on a ViT with DinoV2 weights.

**Quantitative and qualitative evaluation**   Table 2 shows the quantitative evaluation of KCC and numerous SEM baselines. In this case, a ViT-based model with pretrained weights from DinoV2 (Oquab et al., 2024) are used. Due to architectural constraints and training vs. no-training, a direct comparison is challenging. However, the results show that training-free methods can achieve comparable or better performance to supervised SEMs for some datasets. For example, KCC has better accuracy than ProtoPNet on CUB200, despite not being trained on this dataset at all. The most direct comparison for KCCs is KMEx, which achieved superior performance but at the cost of worse explainability, as shown in Table 1. Also note that for the CARS dataset, the training-free methods struggle, indicating that for fine-grained classification with highly specific features, supervision is still necessary.

**Towards reduction of reader bias with automatic text description of keypoints**   A universal challenge for all XAI methods and their visualization is that the reader must interpret the explanations. This introduces a reader bias (Bove et al., 2024), since users can interpret the visualizations differently. Here, we show how KCCs can reduce this reader bias by taking advantage of recent ViTs with vision-language capabilities (Sun et al., 2023). The recent VLParts model (Sun et al., 2023) offers a general purpose foundation model that segments images into parts, where each part is automatically labeled with a text description from an open vocabulary. This removes the ambiguity in the interpretation of the visualizations and explicitly states what is being matched. We replace the part segmentation model of a KCC with the VLParts and follow the general methodology of KCCs.

Figure 1 shows how the introduction of ViT foundation models with vision-language capabilities can add an additional benefit to KCCs, namely automatic labeling of the keypoints. This removes the reader bias since the meaning of the keypoints is now explicit. We also evaluate the performance of this approach on the CUB200 dataset which achieves an accuracy of 82.7 and a complexity of 3, thus comparable performance to models in Table 2. We believe that the combination of KCCs and vision-language modeling offers a promising direction forwards for reducing reader bias in XAI.

## 6   DISCUSSION

**Equal weighting of keypoints**   A key element of KCCs is that each keypoint is weighted equally, with the motivation that the reader only needs to be able to count to understand the explanation. However, it is evident that there will be keypoints with higher importance to a particular class, for example a highly class-specific beak shape in the bird classification setting. A straight-forward way to incorporate such information could be to weight the keypoints e.g. by the distance between keypoints. However, we deliberately avoid this as it will make understanding the decision significantly less intuitive, since the user must asses an arbitrary weighting factor without an inherent meaning. An alternative approach would be to identify class-specific keypoints and remove keypoints that

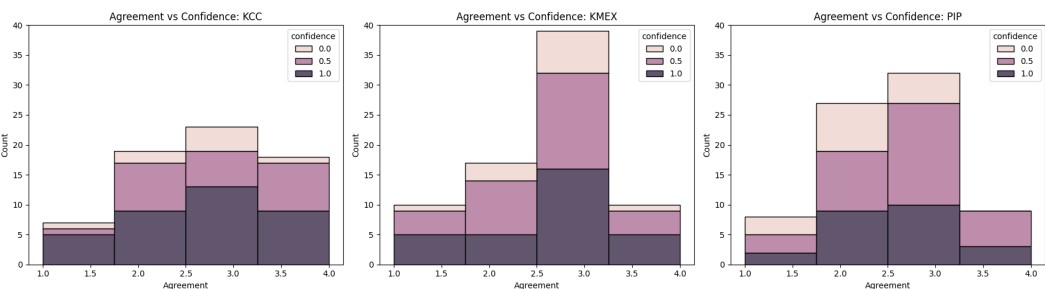

Figure 4: Results from user study on user confidence as a function of agreement with explanation. The results show that KCCs allows people to be more confident in correcting the model.

are shared among many classes to remove redundant keypoints. However, such a procedure is not straight-forward and we envision this as an interesting line of future research.

**Findings from user study** The results of the user study in Table 1 shows many interesting findings. User preferences is important since it indicates that users feel comfortable working with the explanations. But the user preference must always be seen in light of the quantitative measurements, since an algorithm can have a high user friendliness but still lead the user to make wrong decision. In this regards, KCC is superior to KMEx, where a high user preference is combined with improved quality and understanding of explanations.

Another important aspect is the issue of automation bias (Skitka et al., 1999), in which human have a tendency to trust the predictions of automatic systems. This automation bias is visible for all methods in Table 1, where participant struggle to identify incorrectly classified samples, which is in line with prior studies Kim et al. (2022). However, the participants were also asked; "do you feel confident to make the right prediction based on the explanation shown"?, which is not shown in Table 1. This is important, because a step towards reducing automation bias is enabling users to confidently agree/disagree with the model. Figure 4 shows the confidence score of the user as a function of agreement with the model. When users disagree with a prediction, they select an agreement of 1 or 2, and in those cases, KCC has higher confidence scores compared to KMEx and PIP. This indicates that KCCs allows people to be more confident in correcting the model.

## 7 CONCLUSION

We present a new paradigm for self-explainable deep learning that requires no training, is suitable for ViTs, and provide a new way of visualizing explanations. A user study showed that KCCs allowed humans to understand the explanations to a greater extent compared to existing approaches. We also showed how exploiting recent advances in vision-language modeling could provide automatic text description towards reducing reader bias. We believe that KKCs provides a completely new direction within XAI with great potential to improve the transparency of deep learning.

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

# A QUALITATIVE EXAMPLES

Two supplementary examples of KCCs with automatic text descriptions are presented to illustrate the generalizability of our results.

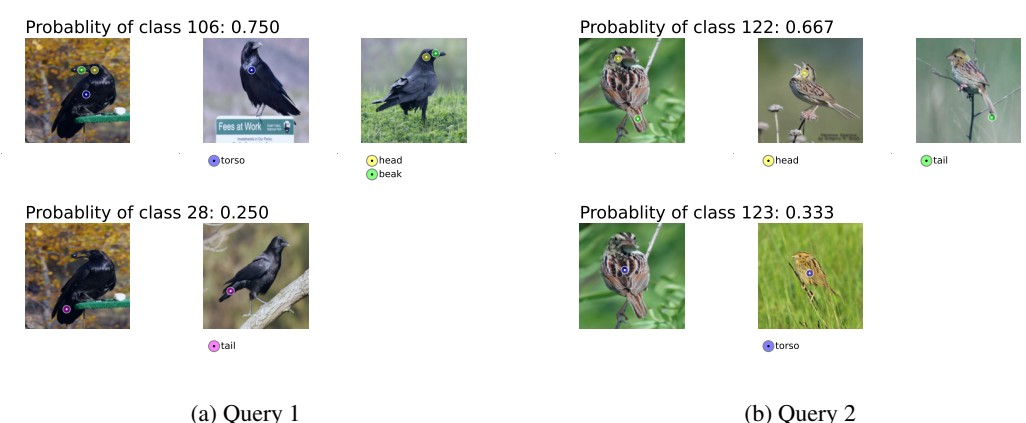

(a) Query 1                    (b) Query 2

Figure 5: Automated keypoints description using vision-language model

# B HYPERPARAMETERS

This section shows results for different number of segments, different number of prototypes, and with different encoder

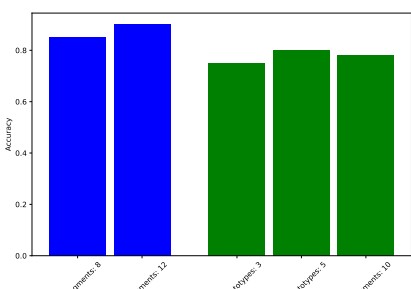

Figure 6: Different number of segments and prototypes.

| method | encoder | CUB200 | CARS | PETS |
|--------|---------|--------|------|------|
| KMEx | clip | 61.0 | 68.3 | 75.0 |
| KCCs (ours) | clip | 57.6 | 64.0 | 75.4 |

Table 3: KCC with CLIP encoder.

