# OpenReview forum: "Keypoint Counting Classifiers: Turning Vision Transformers into Self-Explainable Models Without Training"
_ICLR.cc/2026/Conference — ICLR 2026 Conference Withdrawn Submission_

### Official Review · Reviewer_3ixq · 2025-10-24

**Soundness:** 1
**Presentation:** 2
**Contribution:** 1
**Rating:** 2
**Confidence:** 4

**Summary:**

The paper introduces a novel, training-free method for creating self-explainable models from pretrained Vision Transformers (ViTs). The core contribution lies in leveraging keypoints to generate explanations without the need for additional training. The authors evaluate their proposed method's accuracy and complexity on three datasets, comparing it against both training-free and non-training-free baselines. Furthermore, the paper presents a user study that assesses the explanation quality and user understanding of the proposed method in comparison to bounding box and heatmap-based approaches. The results of this study indicate a slight user preference for the proposed method.

**Strengths:**

* **S1 (Tackling an Important Problem)**:  The paper focuses on making ML models easier to understand by creating models that can explain their decisions without needing extra training.

* **S2 (Novel Idea)**: The method relies on existing methodologies, however, using keypoints to make a ViT model a SEM is a new approach.

**Weaknesses:**

* **W1 (Clarity of the Methodological Description)**: The explanation of the proposed method could be enhanced to improve readability and reproducibility. The notation in the methodology section is at times inconsistent or undefined. For instance, in Equation 1, S(j) and Z are used without a clear definition. A more detailed and systematic explanation of the mathematical formulations would greatly benefit the paper.

* **W2 (Details of the Implementation)**: In lines 164-166, the paper states that foreground segmentation is performed using Segment Anything and Grounding DINO. Additionally, the paper states that ViT is used to automatically describe the keypoints. Could the authors please provide a more detailed explanation of how this is accomplished, as this information appears to be missing from the current draft.

* **W3 (Ambiguity in Prototype Selection)**: The process for selecting prototypes is not clearly described. I suppose they are randomly chosen from the dataset but the target class must be included otherwise the keypoint match would provide a false match. A more thorough elaboration on this crucial step would improve the paper's completeness.

* **W4 (Limited Experimental Evaluation)**: The experimental setup appears somewhat limited. The evaluation could be made more robust by including experiments using standard benchmarck datasets such as CIFAR100 and ImageNet1k and by exploring the impact of different ViT architectures and sizes. Additionally, Table 2, which reports accuracy, does not include standard deviations, making it difficult to assess the statistical significance of the results.

* **W5 (Comparison to Baselines)**: One of the training-free baselines, KMEx, appears to perform comparably or even better than the proposed method, also when using a different encoder as reported in the appendix. The paper would benefit from a more detailed discussion of the advantages of the proposed method over existing approaches like KMEx, especially in scenarios where the latter shows strong performance. It would be helpful for the authors to elaborate on the unique benefits their method offers.

* **W6 (No code available)**: In the abstract it is stated that the code is available, but it was not provided with the submission.

**Questions:**

* **Q1**: The final prediction is determined by counting matches between query keypoints and prototypes. How does this compare to the original ViT's final prediction? Is the keypoint counting method as accurate? It would also be insightful to see a comparison between the explanations generated by keypoints and the attention maps from the original ViT model, as attention is a common method for interpreting Vision Transformers.

* **Q3**: The paper mentions that ViT itself can be used for this task. It would be beneficial if the authors could provide empirical results for this alternative to substantiate this claim.

* **Q4**: The experiments are focused on a specific ViT model. Could the authors clarify which ViT model was used? Furthermore, have the authors considered evaluating their method on other transformer-based architectures that also use a patch-based approach? Additional experiments with different ViT sizes (e.g., ViT-Base, ViT-Large) would also help to demonstrate the method's robustness.

* **Q5**: To what extent does the proposed method generalize to diverse datasets? It would be valuable to see experiments on standard natural image datasets such as CIFAR-100 or ImageNet-1k.

* **Q6**: In line 190, it is claimed that the method can speed up inference *when* a large number of prototypes are required. Could the authors provide a specific experimental setting that demonstrates and quantifies this speedup? It would be valuable to understand in which cases a *large* number of prototypes and define what large means.

* **Q7**: Regarding the ablation study in Figure 6 of the appendix, a more comprehensive analysis using large numbers would be beneficial. Additionally, could the authors provide a discussion of why performance seems to improve when going from 3 to 5 prototypes but then drops with 10? The rightmost column in the plot is labeled as "# of segments: 10" but is grouped with the prototypes; is this a typo? A brief paragraph explaining the insights from the plots and tables in the appendix would greatly improve their impact.

---

### Official Review · Reviewer_7ZVa · 2025-10-30

**Soundness:** 2
**Presentation:** 1
**Contribution:** 2
**Rating:** 2
**Confidence:** 5

**Summary:**

The work introduces a key concept matching explanations based on DINOv2 model (ViT). It is an alternative for prototypical-parts based model that should provide more intuitive and clear explanations. At the same time the model does not requires retraining and additional fine-tuning. It is supposed to be easy to use and straightforward. The work compares itself to prototypical-parts-based models such as PIPNet, ProtoPool and ProtoS-ViT on CUB, PETS and CARS. Also a user study is performed.

**Strengths:**

The idea is interesting and the approach novel with small disclaimer about usage in this context textual information [8] and correspondence to object parts [2]. I would see it more as generalization of those ideas, which is still valid and novel contribution. However, not groundbreaking.

The introduction is clearly written. Other parts a bit less, see weaknesses.

The paper tackles important topic and is significant for SEM models.

**Weaknesses:**

"despite the fact that many studies have pointed out limitations associated with both bounding boxes and heatmaps" - none of the study showcasing that are referenced. It is not detailed whether heatmaps of SEMs are critized or heatmaps in general. Statements are too big and too vague.

"To improve the usability of SEMs, new methods for visualizing explanations must be developed", work on the improvements of such models exist [1], [2], [3] and should be referenced. This sentence makes the reader think that those attempts have not been done yet, which is not true.

Eq. 2 and its explanation is not clear. A decent figure presenting this concept is missing, as right now text assumes too detailed knowledge about mutual NNs.

Citations are not consistent. Sometimes in brackets sometimes without them.

Typo: "reudcing"

User study is a part of evaluation and should be combined in contributions with evaluation together.

Related works section is chaotic and missing references to recent general purpose SEMs such as QSENN [4] and InfoDisent [5]. And generalization to ViT such as ProtoViT [6] and ProtoPFormer [7] are missing.

Grounding DINO is used for object detection, that produces bounding boxes. How grounding DINO transforms those into the segmentation map is relevant for the work.

In Section 3.1 starting with paragraph "Second", I find it very difficult to follow the work and technical contribution there. The description mixes intuition with operations and subjective opinions and references to literature, which is difficult to fully understand the algorithm and what given symbols mean.

Based on the Figure 2 I do not understand what a prototype is. And how can we tell that something looks like something. So I do not understand if prototype 2 contains 2 keypoints, or each of those keypoints is a prototypical part. Moreover, what is the explanation here? Based on the Figure 1 I don't know what is what, what is input, what is the explanation. How does it corresponds? How does probability connects to keypoints?

Regarding the evaluation, it is not extensive. The purity (Nauta et al., 2023) metric is missing. Local and global size of explanation is missing (only complexity is provided). Computational experiments are firstly presented in Section 4 and in Section 5 they are later which is odd and misleading. Baselines such as InfoDisent, ProtoViT and ProtoPFormer are missing. I don't know for which number of segments results in Table 2 are presented. As I understand, the user look up to 2 prototype per image for explanation which is a strong limitation because as it is provided in (Rymarczyk et al., 2022) from cognitive point of vie the optimal number of concept to shows is between 4 and 9. This should be highlighted. Lack of deviation in the accuracy results cannot tell us about the variability of the model. Also original results of the ViT methods on other backbones should be reported and ablation how backbone influences the results is missing - the paper claims it is backbone invariant, so should be more backbones provided to be able to compare with ProtoViT and InfoDisent.

When it comes to quantization of explainability, lack of experiments on FunnyBirds [9], misalignement benchmark [3] and proper purity, local and global sizes (Nauta et al., 2023), are making this work difficult to compare and assess its advantages over other methods.

Details on user study are missing, which platform, how does look like the age and sex of the participants. Are they across the world or a group of students? What is their background? What measures have been taken to check whether the answers are correct? How was the user study made randomized? Also, follow-up works [1, 2] have conducted HIVE-based user studies and they should be compared to.

When it comes to description of keypoints, this should be compared to [8], as this kind of feature is already within the literature and the idea is not exactly novel.

There is no discussion whether this can be applied outside fine-grained recognition, e.g. in ImageNet, as other approaches such as [5] allows it.

Also there is lack of ablation on certain components and hyperparameters of the model.

[1] Ma, Chiyu, et al. "This looks like those: Illuminating prototypical concepts using multiple visualizations." Advances in Neural Information Processing Systems 36 (2023): 39212-39235.
[2] Pach, Mateusz, et al. "LucidPPN: Unambiguous prototypical parts network for user-centric interpretable computer vision." ICLR (2025).
[3] Sacha, Mikołaj, et al. "Interpretability benchmark for evaluating spatial misalignment of prototypical parts explanations." Proceedings of the AAAI Conference on Artificial Intelligence. Vol. 38. No. 19. 2024.
[4] Norrenbrock, Thomas, Marco Rudolph, and Bodo Rosenhahn. "Q-senn: Quantized self-explaining neural networks." Proceedings of the AAAI Conference on Artificial Intelligence. Vol. 38. No. 19. 2024.
[5] Struski, Łukasz, Dawid Rymarczyk, and Jacek Tabor. "Infodisent: Explainability of image classification models by information disentanglement." arXiv preprint arXiv:2409.10329 (2024).
[6] Ma, Chiyu, et al. "Interpretable image classification with adaptive prototype-based vision transformers." Advances in Neural Information Processing Systems 37 (2024): 41447-41493.
[7] Xue, Mengqi, et al. "Protopformer: Concentrating on prototypical parts in vision transformers for interpretable image recognition." arXiv preprint arXiv:2208.10431 (2022).
[8] Wan, Qiyang, Ruiping Wang, and Xilin Chen. "Interpretable object recognition by semantic prototype analysis." Proceedings of the IEEE/CVF Winter Conference on Applications of Computer Vision. 2024.
[9] Hesse, Robin, Simone Schaub-Meyer, and Stefan Roth. "Funnybirds: A synthetic vision dataset for a part-based analysis of explainable ai methods." Proceedings of the IEEE/CVF International Conference on Computer Vision. 2023.

**Questions:**

With regards to my score, to increase it from 2 to 4 I would like to ask authors to:

- provide necessary details related to user study, please provide the same level of details as in [1,2]. I am not convinced that it has been done properly. If those details will not satisfy the current standard I will not increase my score.

- improved evaluation: provided missing metrics such as local, global sizes, purity and evaluation on spatial misalignement and FunnyBirds

- improved results reporting: adding variation to the accuracy values

- missing comparisons added, especially with Q-SENN, ProtoViT, InfoDisent and ProtoPFormer

- ablation on ViT backbones that truly showcases the versatility of the approach

- better phrasing of contributions with recent literature in mind

To increase to 6:

- all of the above plus work on the clarity, especially methodological description

- better visualization of explanations, more examples

To increase to 8:

- better visualization and figures,

- discussion of limitations and impact of this work, as well as ethic and reproducibility statements

- adding missing literature and discussion to Related Works

**Details Of Ethics Concerns:**

There is a user study performed. No details on participants are provided and their background.

---

### Official Review · Reviewer_3bQ6 · 2025-10-31

**Soundness:** 2
**Presentation:** 2
**Contribution:** 2
**Rating:** 2
**Confidence:** 3

**Summary:**

The paper introduces Keypoint Counting Classifiers (KCCs), a post-hoc method to make ViTs self-explainable without requiring any additional training or fine-tuning. Current SEMs are limited to specific architectures like CNNs, making them incompatible with ViTs, and employ visualizations through bounding boxes or heat maps, which can be misleading or difficult to interpret. KCCs are proposed as a training-free method to address the gap in explainability for ViTs that also use a more intuitive visualization than bounding boxes or heat maps. The method first performs foreground segmentation with SAM and GroundingDINO to identify the foreground object and then uses SLIC to divide it into semantically coherent segments, defining a keypoint at the center of each segment. It then finds matching keypoints between the query and prototype images by computing mutual nearest neighbors over the query and prototype tokens. Finally, the model makes its decision by counting the number of matching keypoints for each class.

**Strengths:**

- The goal of introducing a training-free self-explainability method for ViTs is novel and directly addresses the inflexibility of prior SEMs that require specific architectures or costly retraining.
- The user study using different metrics for comparing user experiences between different explainability methods was well-designed.
- The paper is clearly written and the method description is easy to follow.

**Weaknesses:**

- The paper's goal is to create a self-explainable method where the decision process is "inherently explainable", however it relies on external, non-explicitly explainable segmentation models (SAM and Grounding DINO) at the very beginning of the self-explanation pipeline for foreground segmentation. The work would benefit from an argument as to why additional non-explicitly explainable models can be used as part of the self-explanation pipeline.
- As noted in Section 4, this method quickly becomes computationally expensive as the number of classes, prototypes, and number of tokens in the query and prototypes increase. The work would benefit from a discussion on the scalability of the method.
- Only a handful of quantitative results in Table 1 are statistically significant (bolded) making it difficult to draw any meaningful conclusions about the relative performance between the methods. The quality and understanding metrics across all methods all have overlapping standard deviation intervals and KMEx and KCC achieve the exact same user preference score. Further, according to Table 2, KMEx seems to be more accurate than KCC for every dataset tested.
- The paper's quantitative evaluation in Table 2 includes the CUB200, Stanford Cars, and Oxford Pets datasets, but only images from the CUB200 dataset are used in figures, so it is unclear how well the method's visualizations perform on other domains. The work would benefit from more diverse qualitative examples as well as a discussion on common failure modes, e.g. why training-free methods struggle on CARS.

**Questions:**

Please address the concerns outlined in the weaknesses. Additionally, how is the parameter J representing the number of closest prototypes being decided, and where are the results for different values of J?

---

### Official Review · Reviewer_Kjpc · 2025-10-31

**Soundness:** 2
**Presentation:** 3
**Contribution:** 2
**Rating:** 4
**Confidence:** 3

**Summary:**

This paper introduces Keypoint Counting Classifiers (KCCs), a novel framework for converting pre-trained Vision Transformers (ViTs) into self-explainable models without requiring any retraining. The method operates by identifying semantic keypoints on objects (using external models), matching them between a query image and a set of class prototypes using mutual nearest neighbors, and performing classification by counting the number of successful matches per class. The primary contributions include the training-free nature of the method, a new keypoint-based visualization for explanations, and a comprehensive user study validating its effectiveness in improving human-AI communication.

**Strengths:**

- The paper introduces a training-free method for generating post-hoc explanations for ViT models. The proposed keypoint-based visualization offers an alternative to common explanation modalities like heatmaps, making the work relevant to ongoing efforts in XAI.
- The authors conduct a user study to compare their proposed visualization against baseline methods, and the results suggest a user preference for their approach in terms of perceived quality. The quantitative analysis benchmarks the method's accuracy on several fine-grained classification datasets.
- The paper is adequately structured and the core methodology is presented in a way that is generally easy to follow.

**Weaknesses:**

1. **Contradiction in "Self-Explainability" and Lack of Faithfulness:** A major conceptual weakness arises from the reliance on external models (SAM, Grounding DINO). By forcing a foreground mask, the method is not explaining the original ViT's decision but rather a constrained decision within a pre-processed input. This compromises the faithfulness of the explanation, as it cannot reveal if the model relies on spurious background correlations for its prediction. Furthermore, depending on external models to generate the explanation space contradicts the principle of a "self-explainable" system, which should ideally derive its explanation from its own internal representations and logic.

1. **Unsupported Generality of the "Any ViT" Claim:** The claim that KCC can turn "any" ViT into an SEM is not fully substantiated. The experiments are conducted exclusively on ViTs with DinoV2 weights, which are known to produce highly structured patch tokens ideal for correspondence tasks. The method's efficacy on standard, supervised ViTs (e.g., trained on ImageNet-1K for classification) is not demonstrated. These models may lack the requisite feature consistency for the matching mechanism to succeed, potentially limiting the method's applicability to a specific class of self-supervised models.
2. **Scalability to Large-Scale Datasets:** The method's scalability to large-scale datasets is a significant practical concern. The inference-time process of matching a query against a large number of prototypes (e.g., 10 prototypes for 1,000 classes in ImageNet) appears computationally prohibitive in terms of both memory and latency. The paper should address the practical feasibility of this approach by providing computational cost analysis based on the number of classes.
3. **Missing Implementation Details:** The paper omits several implementation details crucial for reproducibility. It fails to specify: (1) which layer of the ViT is used to extract features for keypoint matching, (2) the version or size of the ViT backbone used for DinoV2 (e.g.ViT-L, ViT-G…, with/without registers?), and (3) the specific CLIP model variant used in the appendix Table 3 (which is not discussed anywhere in the paper). These choices can significantly impact performance and feature quality.

**Questions:**

1. Performance on Supervised ViTs: Could the authors provide results (accuracy and qualitative explanation quality) of KCC's performance when applied to a standard ViT trained with a supervised classification objective on ImageNet? A negative result would be highly informative and would help clarify the true scope of the method's applicability.
2. Addressing the Foreground Constraint: Regarding the reliance on foreground segmentation: Have the authors considered an alternative where keypoints can be generated from the entire image? While potentially noisier, this would allow the method to identify if the model is using background features, thereby improving the faithfulness of the explanation.
3. Analysis of Computational Cost: Could the authors provide a more detailed analysis of the computational cost? Specifically, what is the inference latency and memory footprint for a dataset like CUB-200, and how is this projected to scale with an order-of-magnitude increase in classes and prototypes?
4. Clarification for Reproducibility: For reproducibility, could the authors please clarify which specific layer of the ViT was used to extract the patch token features? It is assumed to be the final layer, but explicit confirmation would be beneficial.

---

### Author Response · Authors · 2025-11-12
**Thank you for the feedback on our manuscript.**

Dear reviewers,

Thank you for the valuable feedback on our manuscript. We believe that using keypoint counting as a mechanism to turn ViT's into a SEM is an interesting research direction with great potential. But we appreciate the limitations that the reviewers have raised, and we agreed there is room for improvement across several aspects of our work. Therefore, we have decided to withdraw the submission. We will take this feedback and improve our manuscript for a future submission. Thanks again for your time and efforts in evaluating our work.

---

### Note · Authors · 2025-11-12

I have read and agree with the venue's withdrawal policy on behalf of myself and my co-authors.